# To address surface reaction network complexity using scaling relations machine learning and DFT calculations

Zachary W. Ulissi[1,*], Andrew J. Medford[2,*], Thomas Bligaard[3] & Jens K. Nørskov[1]

Surface reaction networks involving hydrocarbons exhibit enormous complexity with thousands of species and reactions for all but the very simplest of chemistries. We present a framework for optimization under uncertainty for heterogeneous catalysis reaction networks using surrogate models that are trained on the fly. The surrogate model is constructed by teaching a Gaussian process adsorption energies based on group additivity fingerprints, combined with transition-state scaling relations and a simple classifier for determining the rate-limiting step. The surrogate model is iteratively used to predict the most important reaction step to be calculated explicitly with computationally demanding electronic structure theory. Applying these methods to the reaction of syngas on rhodium(111), we identify the most likely reaction mechanism. Propagating uncertainty throughout this process yields the likelihood that the final mechanism is complete given measurements on only a subset of the entire network and uncertainty in the underlying density functional theory calculations.

[1] SUNCAT Center for Interface Science and Catalysis, Department of Chemical Engineering, Stanford University, Stanford, California 94305, USA. [2] School of Chemical and Biomolecular Engineering, Georgia Institute of Technology, Atlanta, Georgia 30332, USA. [3] SUNCAT Center for Interface Science and Catalysis, SLAC National Accelerator Laboratory, Menlo Park, California 94025, USA. * These authors contributed equally to this work. Correspondence and requests for materials should be addressed to T.B. (email: bligaard@slac.stanford.edu) or to J.K.N. (email: norskov@stanford.edu).

Reaction network complexity limits the understanding and modelling of experimental behaviour in combustion, metabolic engineering and catalysis, among other fields. The sheer number of possible intermediates leads to reaction networks with hundreds or thousands of species, thousands of reactions and an exponential number of possible pathways and mechanisms to be considered. In all of these fields, studying individual reactions is a costly and time-consuming process. For direct hydrocarbon reactions in combustion and catalysis, density functional theory (DFT) allows for estimation of kinetic reaction parameters with a reasonable degree of accuracy, but at significant computational cost. In metabolic engineering, there is no straightforward method to estimate accurately enzyme kinetics, and most kinetic parameters are derived from experimental studies. Identifying the right reactions to focus computational and experimental resources on is thus of paramount importance. This problem is especially difficult in catalysis since the reaction network varies greatly across different catalyst surfaces and active sites, and thus the precise mechanism must be reidentified for every catalyst, as illustrated in Fig. 1 for the reaction of syngas over Rh(111). This process is a fundamental limitation to the design of new catalysts and introduces error in the interpretation of experimental data if the wrong mechanism is derived.

The overwhelming complexity of reaction networks can be addressed with the insight that most of the network has little impact on the final results and can thus be treated with less accurate surrogate models. The most important properties of these reaction networks, the kinetic parameters of the rate-limiting step, exhibit little sensitivity to the majority of the reaction network. Model refinement should focus on the most important reactions. This approach is fundamentally similar to insights from sloppy modelling in the systems biology community, where sensitivity analyses show that refinement in some model parameters will have a limited impact on key system observables[1] and allowing low-quality estimates to be used. Using surrogate models as a guide for full-accuracy electronic structure calculations allows for a rapid exploration of new reaction networks. Similar approaches are used in related fields. The combustion literature has developed approaches to automate the study and reduction of gas-phase kinetics using group additivity energetics and automated semiempirical quantum mechanical calculations[2]. Metabolic engineering has also begun to focus on reaction mechanism reduction techniques, now that well-annotated genome-wide data sets are available[3]. In heterogeneous catalysis, it is widely accepted that adsorption energetics for various species are fundamentally related on the same surface and change in predictable fashions when moving to other surfaces, resulting in broadly applicable linear scaling relations[4]. There has also been success in applying detailed group-additivity approaches to the study of large reaction networks[5–10] and identifying the impact of uncertainty on these processes[11,12]. Previous work has also used precalculated semiempirical models to aid in model reduction[13]. As these methods have become increasingly accurate they have focused on developing approximations that can completely predict the properties of a reaction network but with increased complexity and cost.

This work takes a markedly different approach and is distinguished by two key insights. First, we cast the problem of surface reaction mechanism elucidation in a fully probabilistic framework, starting for a surface for which no mechanism or surface energetics are available. This approach provides a route to bootstrap to a mechanism of optimal complexity rather than relying on intuition or expensive comprehensive analysis. Tracking the uncertainty at various levels of approximation enables the use of low accuracy methods for most of the reaction

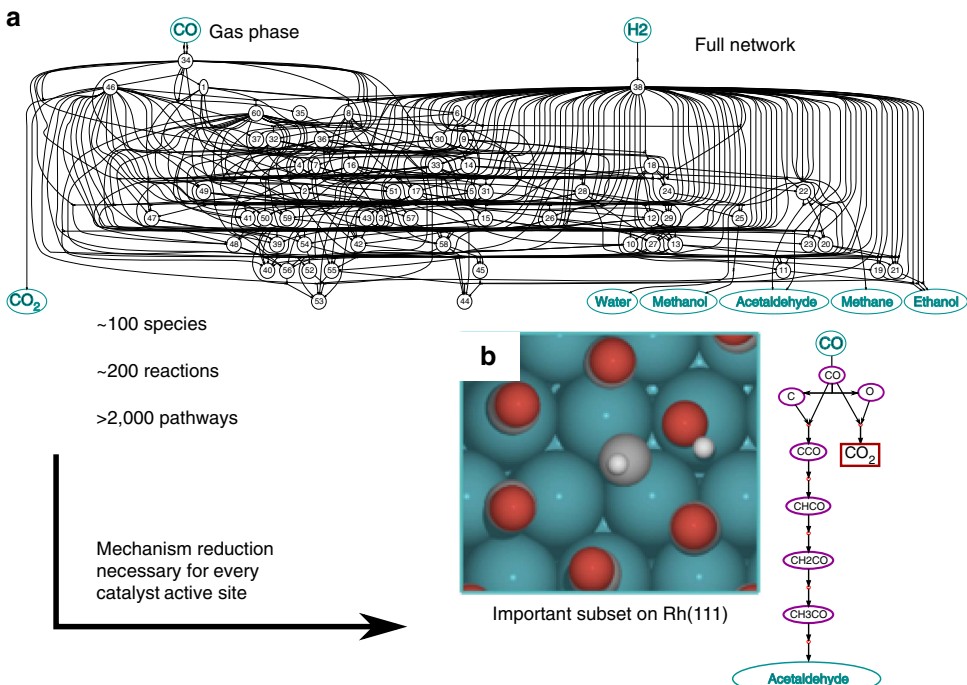

**Figure 1 | The challenge of reaction network complexity in catalyst discovery.** (**a**) Reaction network for the reaction of syngas (CO + H$_2$) to CO$_2$, water, methane, methanol, acetaldehyde and ethanol, including surface-adsorbed intermediates with up two carbons and two oxygens (C1/C2 chemistries). Even for this reduced network, there are hundreds of reactions and thousands of possible mechanisms to consider for each new catalyst active site, which are prohibitively expensive for materials discovery screens. (**b**) The reduced network for syngas reactivity on Rh(111), producing acetaldehyde selectively as confirmed by the experiment. The reduction from (**a**) to this subset is made more efficient and uses far fewer full-accuracy DFT calculations using the methods in this work.

networks and builds on the recent introduction of uncertainty-enabled DFT calculations. Second, this approach is more approachable to the broader chemistry community than existing methods due to its design and simplicity. We place a large premium on being able to quickly identify the rate-limiting steps so that we can engineer those reactions with modifications to the underlying catalyst without a sophisticated model. The fact that a remarkably simple model can capture the most important underlying physics is precisely what makes them so powerful. In all ways, this workflow represents how a typical researcher in our field would approach the problem of studying a new surface, except that the intuition for choosing the next reaction to study is replaced by simple group-additivity-based methods that are more accurate than a human at guessing the energetics for unstudied reactions.

In this work, we focus accurate but computationally expensive DFT methods on the reactions that are likely to be the rate-limiting steps in the reaction network. Properties of the rest of the network are inferred using a surrogate model based on physics-based approximations that have already been established for understanding catalyst trends, such as linear scaling relations, and the accuracy of these methods is quantified. Propagating uncertainty at every layer of approximation allows for estimates on the residual error in the final reaction mechanism from parts of the reaction network that have not been studied in detail. These methods also avoid wasting computational resources on reactions that are part of the final reduced mechanism but not likely to be the rate-limiting step (for example, fast hydrogenation reactions). This process is robust to the accuracy of the approximations used since any important reactions are studied with full-accuracy DFT calculations and added to model training sets. Starting from a few DFT calculations and iterating this approach generates the most likely pathway with fewer calculations than would be necessary to study the entire network. We demonstrate these ideas for the reaction of syngas ($CO + H_2$) on Rh(111), a reaction that has been studied both experimentally and computationally[14], and demonstrate reductions of 60% and 95% for the number of intermediate and transition-state calculations, respectively. This process is demonstrated for a single set of experimentally relevant thermal conditions (573 K, 1 atm partial pressure of all gas-phase species) but could easily be repeated for other conditions.

## Results

**General**. Modelling catalyst surface chemistry is a multistep process from understanding intermediate adsorption configurations to identifying kinetic reaction barriers, as illustrated in Fig. 2a. DFT is particularly well suited to this problem with a favourable compromise between chemical accuracy and computational resources. Recent advances have also allowed for estimates of the uncertainty in DFT calculations by using an ensemble of parameterizations[15]. Typical uncertainty for the DFT methods used in this work (estimated with the ensembles in Bayesian error estimation functional with van der Waals correlation (BEEF-vdW)) are $\sim 0.15\,eV$ for surface species formation energies[16], and $\sim 0.2$–$0.3\,eV$ for transition-state formation energies (relative to gas-phase species) as indicated in Fig. 2. At every step, we can replace DFT calculations with approximations that draw on an existing training set of calculations or physics-based approximations. For example, we can use a set of DFT calculations of surface intermediate formation energies to train a machine learning regression scheme based on group additivity fingerprints. Although these methods introduce additional uncertainty into predictions for unmeasured quantities, they are sufficiently accurate to exclude parts of the

network that are clearly unfavourable. Standard methods are used for electronic structure of surface intermediates and transition states, as described in the Methods section and the Supplementary Methods.

Parts of the reaction network that are calculated to be rate limiting are studied with full-accuracy DFT and added to the training set to improve predictions for the remainder of the network, as illustrated in Fig. 2b. The process is boot-strapped with a small number of DFT calculations that are likely to be part of the final reaction network, such as the adsorbed species for gas-phase reactants and products, as well as well as elemental binding energies. Properties of full network are predicted, rate-limiting transition states identified and studied with DFT, and these measurements are used to improve the accuracy of future predictions. This process can be broadly classified into a simple mechanism enumeration scheme (described in Supplementary Note 1), a prediction scheme for surface formation energies, a prediction scheme for transition-state energies and a simplified reaction model to identify rate-limiting steps.

**Prediction of reaction kinetics and rate-limiting steps**. A hierarchy of predictive methods are used to provide estimates of transition-state free energies for reaction pathways without relying on the computationally expensive DFT-based CINEB method as outlined in the Methods section. The key approximations are illustrated in Fig. 2a. First, the free energy of each intermediate species is estimated using a combination of machine learning and group additivity methods. These intermediate species energies are then used to calculate reaction free energies for each reaction using stoichiometric relations. Transition-state energies are then predicted using established linear transition-state scaling relations. Finally, significant pathways are determined using approximations from absolute rate theory by tracking the highest transition state in the free energy diagram for each mechanism.

Formation of free energies of intermediate surface species were predicted using a Gaussian process (GP) regression scheme with group-additivity-based fingerprints as illustrated in Supplementary Fig. 1, with details included in Supplementary Note 2. Chemical structures were decomposed into a number of fragments. The number of each type of fragment was fingerprint from which the formation energy could be estimated. Since the contributions of fragments to the formation energy are not linearly independent quantities, as evidenced by the wide applicability of linear scaling relations[4], principal component analysis was used to reduce the dimensionality of this fingerprint to a small number of dimensions (usually about 10–15). A GP was then trained on DFT adsorption energies. As species were selected for study, they were added to the training set and the GP retrained. The free energy of formation and the enthalpy of formation were both predicted in this fashion.

Estimates of transition-state enthalpies from reaction enthalpies were provided with linear scaling relations. Scaling relations between the enthalpy of reaction and enthalpic activation energy were constructed using the CatApp database[17]. Two scaling relations were constructed: one for hydrogenation reactions, and one for all other reactions, as illustrated in Supplementary Fig. 2, the details of which are discussed in Supplementary Note 3. Uncertainty in the final transition-state energy due to the use of these scaling relations was also measured and propagated to the model refinement loop. These transition-state energies were then used to identify the rate-limiting step in the reaction network, as detailed in Supplementary Note 4.

**Model feedback and refinement**. The reaction network model was refined at each iteration by performing DFT calculations on

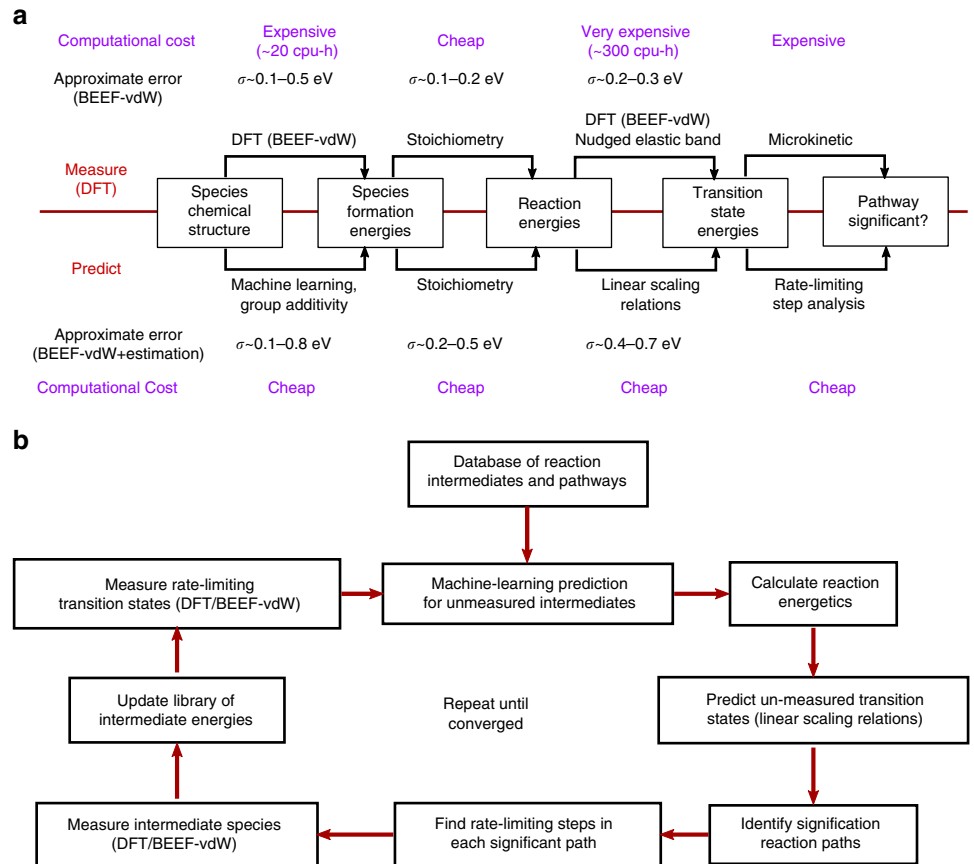

**Figure 2 | Successive approximations and feedback scheme used to determine which reaction pathways are important.** (**a**) Levels of detail necessary to determine whether a reaction pathway is significant. Measurements of each quantity are possible with DFT (and uncertainties provided by the BEEF-vdW functional), but are computationally expensive. These quantities can be predicted from chemical structure using a combination of group additivity, machine learning and linear scaling relations but with greater uncertainty. (**b**) Online model exploration methodology. In each iteration, redictions are made for the transition-state free energy of every reaction. Reactions that have a high likelihood of affecting the most likely pathway are selected for additional study with full accuracy (DFT/BEEF-vdW).

important intermediates or transition states, as illustrated in Fig. 2b. At each iteration, the most likely reaction mechanism was identified from above. For each transition state with at least 10% probability of being the highest-lying transition state, the energetics of reactants and products were measured with DFT, providing a more accurate estimate of the reaction energy and thus the transition-state energy. If the reactants and the products had already been measured with DFT, the transition-state energy was measured with the CINEB method as outlined below. If no species or transition states were chosen for refinement, the next most likely mechanism was chosen instead and this process repeated. This process could be simplified if experimental observations of the apparent transition-state energy or reaction order were available, as these would provide constraints on which reactions were likely to be present in the final model.

The model refinement procedure is visualized in Fig. 3 for the determination of ethanol production mechanism on Rh(111). At the first iteration, very few DFT intermediate energies have been measured and the mechanism predicted by the GP is incorrect, indicating direct scission of CO to elemental C and O. After four iterations, the mechanism is similar except the elemental oxygen is reacted to $CO_2$, and the C and CO both undergo hydrogenations before combination. By the ninth iteration, the correct rate-limiting step has been identified, the scission of CHOH to CH and OH, but there is still uncertainty in the final hydrogenations. By the 22nd iteration, the most likely mechanism has converged and all of the intermediate energetics in the reduced mechanism

studied with DFT. At this point, only 5% of the transition states of the full reaction network have been calculated (all within the final mechanism), and energetics have been calculated for only about 40% of the intermediate species in the full reaction network. In this case, 40% of the reaction network is still quite modest compared to the training sets used to construct group additivity-based models in the past[5,6,11], at the cost of less accuracy. Essentially, the very coarse group-additivity-based model used here is sufficient to exclude clearly unimportant parts of the reaction network, as long as full-precision DFT is used for the remaining important regions, thus retaining full DFT accuracy.

After the selection of the most likely pathway, model refinement focuses on other pathways that are less likely to contain the rate-limiting step. The reduced networks constructed at the 50 or 80% confidence level (that is, the confidence that the final network contains the rate-limiting step at the set of reaction conditions given DFT-level uncertainty) take longer to converge. The 50% confidence level converges to the most likely mechanism, while the 80% confidence level contains additional pathways that cannot be excluded given DFT-level uncertainty.

**Mechanism reduction under DFT uncertainty.** Model uncertainty at the DFT level limits the selection of a mechanism even after all intermediates and transition states have been calculated for a reaction network. Typical uncertainty in transition-state

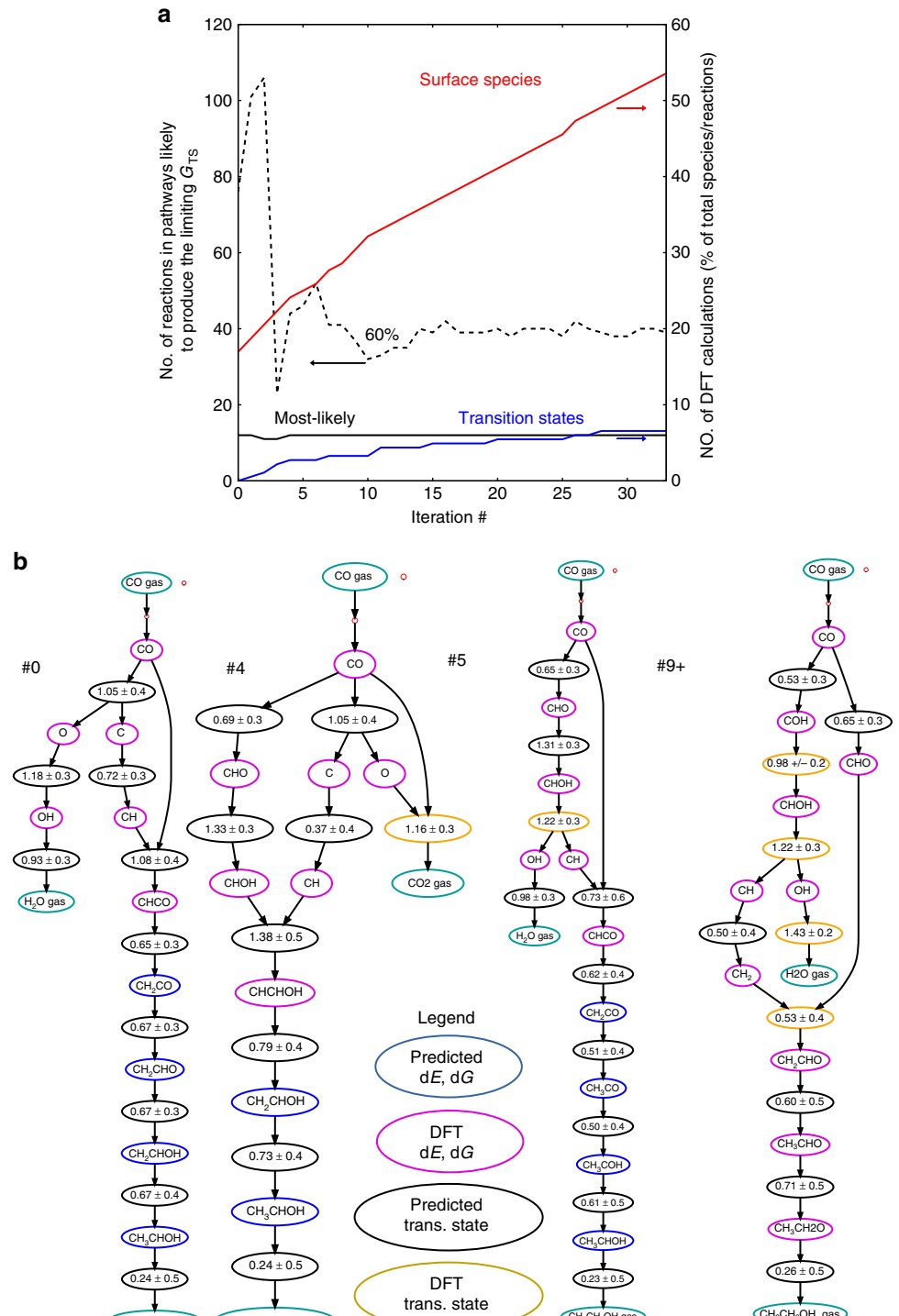

**Figure 3 | Reaction network exploration using predictive reaction energetic models.** (**a**) Convergence of the reaction network at each iteration of the process shown in Fig. 2. At each iteration, DFT calculations are performed for important intermediate species and transition states, allowing model performance to improve at each iteration. (**b**) Convergence of the most likely pathway for the production of ethanol on Rh(111). Direct CO scission is initially predicted to be the most likely process for the scission. By the fifth iteration, the correct CO scission step (CH–OH scission) found in the mechanism is mostly similar to the converged most likely network. By the ninth iteration, the converged most likely pathway is identified, and successive measurements focus on less likely pathways.

energies are typically 0.1–0.4 eV, as estimated using the BEEF-vdW ensemble function. At this level of uncertainty, many pathways through the reaction network may be competitive, as illustrated in Fig. 4. The most likely pathway is shown in Fig. 4b and is consistent with experimentally observed selectivity to

acetaldehyde[14]. The reduced mechanism is included in Supplementary Note 5, along with the complete mechanism in Supplementary Note 6. Including additional pathways that are progressively less likely to contain the network rate-limiting step increases the probability that the the actual transition state is

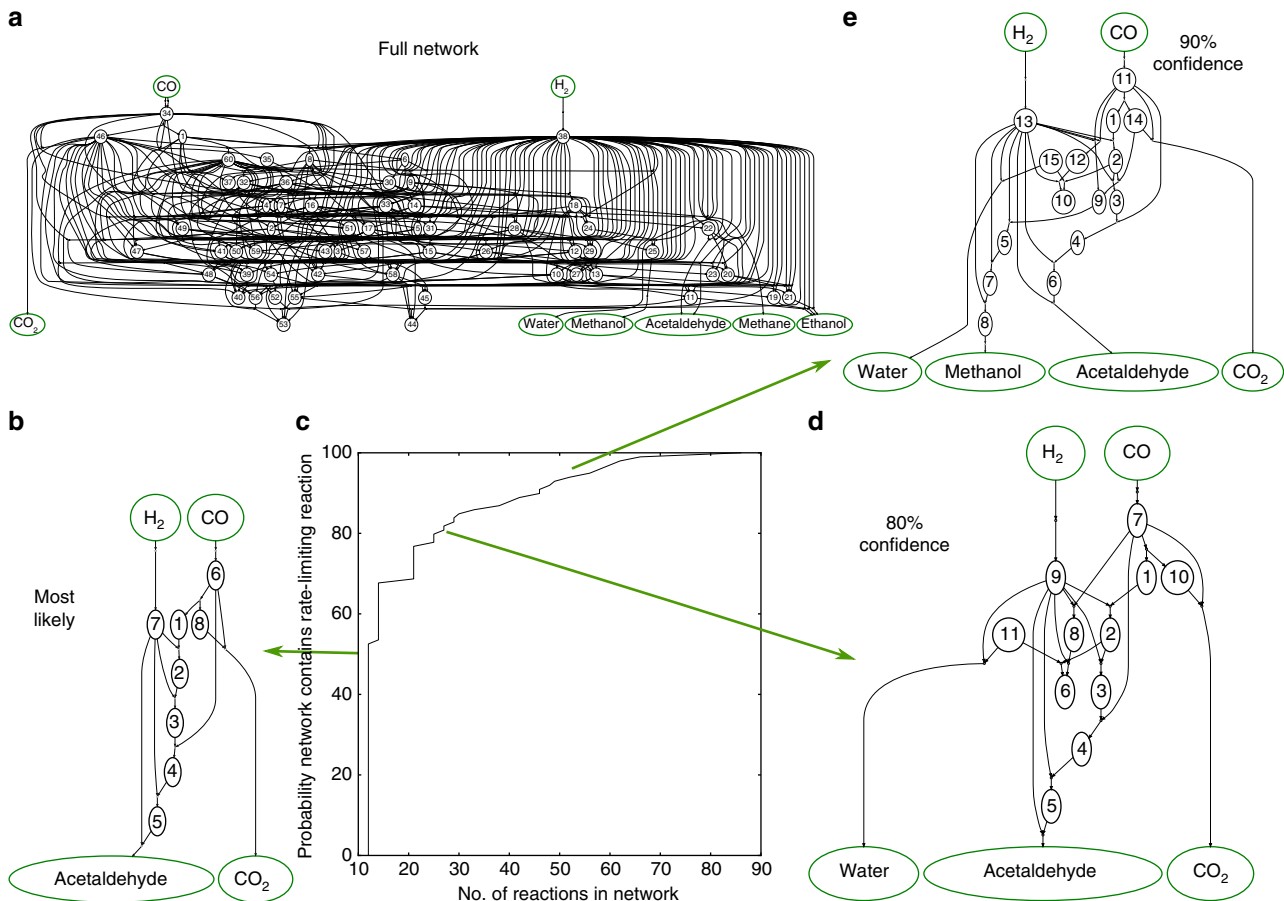

**Figure 4 | Reaction networks for the $CO + H_2$ reaction on Rh(111) under DFT uncertainty provided by the BEEF-vdW functional.** (**a**) All reactions considered for C1 and CCO/OCCO chemistry on Rh(111). (**b**) Pathway with the lowest expected limiting transition-state energy, yielding acetaldehyde as a final product. (**c**) Including less likely reactions in the final mechanism increases the probability that the final network contains the actual rate-limiting step. (**d**) Reduced mechanism at the 80% confidence level, showing selectivity problems to water and $CO_2$. (**e**) Reduced mechanism at the 90% confidence level, suggesting methanol cannot be ruled out as a final product at this confidence level given DFT-level uncertainty.

present, as shown in Fig. 4d. At the 80% certainty level, shown in Fig. 4d, water formation may also be possible. At the 90% confidence level, a desired level of confidence for mechanism reduction, we find that it is impossible to rule out water and methanol as alternate products given DFT-level uncertainty.

The presence of alternate products that compose the network at 90% confidence raises questions about the limitations of using DFT to discriminate between competing reaction networks. Several phenomena might contribute to this problem. First, the uncertainty estimate from the BEEF-vdW ensemble function could be larger than the actual DFT uncertainty. Second, in this work we treat the uncertainty for the linear-scaling relations for transition-state energies to be uncorrelated. However, there could a systematic over- or underprediction for the transition-state energies for a class of reactions that happen to represent the limiting transition states. Testing this is only possible by evaluating the transition state for reaction in the full network and computationally inefficient. However, these estimates will improve as new scaling relations are reported in literature for specific classes of reactions.

## Discussion
Estimating the probability that a given mechanism is correct, given a number of alternative pathways, is only possible when reasonable estimates of model uncertainty are available. For DFT

calculations, this has been a challenge, since various levels of theory and different parameterizations can often lead to varied predictions. The development of ensemble-based approaches to this problem, representing uncertainty that arises from parameterizations of the underlying models fit to sets of experimental data, has already helped. Further work to estimate accurately the uncertainty in DFT calculations will have a large impact in deriving bounds on model predictions for these complex networks. We expect that this work will help remind the community that model error can have a very significant impact on mechanism selection, and that any single mechanism derived solely from DFT calculations should be carefully checked given all of the (large) sources of uncertainty.

Improving the accuracy of surrogate models in this work has a limited benefit in improving the efficiency of network exploration, due to the large separation in pathway energetics between the most and least likely pathways. More accurate methods to predict transition-state energies or species formation energies can be helpful in more quickly establishing the most likely mechanisms. However, we generally require that all species in the final mechanism be studied with full-accuracy DFT for verification, so the accuracy of the surrogate models does not affect the energetics of the final pathway, inspired by surrogate-model approaches in the systems engineering literature[18]. More importantly, there is a trade off in intuition versus simplicity in choosing surrogate models. We desire models that can lead to

physical intuition, such as simplified linear scaling relations in an effort to avoid an overly complex system. We have also focused on approaches that lend themselves to augmenting the existing workflow of computational chemistry (deriving energetics of important pathways), rather than complicated (but more accurate) full systems that aim to control the entire process.

The framework presented here is generalizable to multisite or multicatalyst models as well, given our understanding of how surface adsorption energies on different surfaces are inter-related through linear scaling relations. Implementing this currently requires a separate linear scaling relation for each surface species. We believe linear scaling relations could be fitted on the fly for reaction networks that model activity on multiple surface facets. For example, by including elemental adsorption energies into the fingerprint of each surface species, it may be possible to predict the energetics for arbitrary surfaces given the correct linear scaling relations. Further, some of the intermediate information such as the principal component analysis mapping of molecular fragments are probably reusable from surface to surface as they describe the important fragments that contribute to surface formation energies.

Finally, more accurate modelling of kinetics on the surface to choose the most interesting species to study is desirable. Integration of these methods with existing microkinetic codes such as CatMAP would simplify this process. We have found that the chosen criterion (highest transition-state energy in a pathway) has worked well for determining the rate-limiting step in the networks presented here, likely due to the proximity of the rate-limiting step to the beginning of the pathway. Reaction networks with multiple competing transition states in series would likely be better modelled using a full microkinetic approach.

## Methods

**Computational.** All electronic structure calculations are carried out via the open-source package Quantum ESPRESSO[19]. The exchange-correlation energies are approximated using the BEEF-vdW[15] functional, which uses the generalized gradient approximation and includes a non-local van der Waals correction. All calculations are preformed using a plane-wave basis (plane-wave cutoff of 500 eV, density-wave cutoff of 5,000 eV) and the Brillouin zone is sampled by a Monkhorst-Pack $k$-point mesh[20]. The lattice constant of Rh was calculated with by minimizing the energy of the $1 \times 1 \times 1$ bulk unit cell ($k$-point sampling $12 \times 12 \times 12$). The lattice constant was determined to be 3.838 Å, which compares well with the experimental value of 3.803 Å (ref. 21). Surface calculations were conducted using a standard approach[14], but for completeness these details are also included in the Supplementary Methods.

**Data availability.** The data that support the findings of this study are available from the corresponding author on reasonable request.

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

## Acknowledgements

Support from the US Department of Energy Office of Basic Energy Science to the SUNCAT Center for Interface Science and Catalysis is gratefully acknowledged. A.J.M. was supported by the Department of Defense (DoD) through the National Defense Science and Engineering Graduate Fellowship (NDSEG) Program.

## Author contributions

Z.W.U. implemented the group-additivity prediction scheme and the model refinement and exploration process. A.J.M. performed the DFT calculations and implemented the network enumeration and analysis methods. T.B. aided in developing the approximations and uncertainty propagation methods. J.K.N. conceived the problem. All the authors contributed to writing the paper.

## Additional information

**Competing financial interests:** The authors declare no competing financial interests.

**Publisher's note**: 

