## [Peer Review File · Nature Communications]

REVIEWERS' COMMENTS:

Reviewer #1 (Remarks to the Author):

The authors have addressed most of the comments raised previously and have set this contribution in the context of the broad literature. This work does propose a valuable new approach towards modeling complex reaction networks in a practical way. This work can therefore be published. My only suggestion, in the pedagogical spirit of explaining the inner workings of the method to potential future practitioners, is that the authors explain their PCA and GP more clearly in the SI. I think it is non-trivial to reduce the dimensions of the fingerprint from 50 to 10. Also, it is unclear how the GP was trained. The molecule is composed of groups and therefore the fingerprints are vectors of integers- so it is not apparent what the covariance function of the GP is, the justification for it, etc. It would be good to compare the proposed GP method with standard linear regression based group additivity methods, show the performance of this method (say a parity between DFT and GP-predicted values), etc.

Reviewer #2 (Remarks to the Author):

I appreciate the effort of the authors in clarifying their paper. The main concern I had was the novelty of the work compared to previous work. In their rebuttal, the authors state 'Most of the cited work require significant insight into the existing model, usually using pre-existing microkinetic models and energetics.' And they continue 'We show that all of this information can be efficiently explore on-the-fly during mechanism refinement, starting from no known surface energetics.' Reading the SI multiple times, I see that the mechanism is manually generated and the kinetic information relies on the scalings (Eqs. 3-5), all of which are done a priori rather than on the fly. The only aspect that is done on the fly is the thermochemistry using the GP rather than an a priori estimation scheme. The use of PCA is standard (is one of the several schemes to identify the dimensionality of the space) as is model reduction. I don't see why the on the fly thermochemistry is a major advantage or significantly different from prior art. On this basis, I still believe the paper lacks the novelty for the journal and the typical breakthrough work we're used to see from this group and should be published in a specialized journal.

Reviewer 1

General remarks:

1. *My only suggestion, in the pedagogical spirit of explaining the inner workings of the method to potential future practitioners, is that the authors explain their PCA and GP more clearly in the SI. I think it is non-trivial to reduce the dimensions of the fingerprint from 50 to 10.*
 - a. We appreciate this attention to detail, and have included more information in the Supplementary Information so that these results can be easily recreated. For the PCA reduction, we agree that it is difficult to reduce 50 dimensions to 10 for large well-distributed datasets. However, the fingerprint vectors of chemical groups are highly correlated. The dimensionality reduction of the PCA thus represents correlations in how groups are added and removed to form different intermediate species. For example, adding a –OH group to a carbon is always accompanied by a corresponding fingerprint for the –H attached to the oxygen and there is no additional information gained by having this change reflected twice in the fingerprint vector. The dimension of the fingerprint corresponding to the hydroxyl hydrogen is thus meaningless. Most of the principle components contain a feature related to a larger carbon or oxygen residue and a number of associated hydrogens that appear when that residue is introduced; that is, a vector that represents the addition of a new subgroup of importance. We choose PCA because it addresses the issue of correlation in the subspace without measurements (it is an unsupervised technique).
2. *Also, it is unclear how the GP was trained. The molecule is composed of groups and therefore the fingerprints are vectors of integers- so it is not apparent what the covariance function of the GP is, the justification for it, etc.*
 - a. We have included details about the GP, correlation, and parameters in Supplementary Note 2, as well as how these decisions were made. In most cases, they were found empirically to represent the best trade-off between yielding useful predictions and reproducing training points exactly.
3. *It would be good to compare the proposed GP method with standard linear regression based group additivity methods, show the performance of this method (say a parity between DFT and GP-predicted values), etc.*
 - a. This is an interesting question. A linear regression based approach might not function well because most molecules interact through just a single central binding atom with the surface. Thus, the adsorption free energy is not a sum over contributions of all components, but instead based on the contribution of the component that is most strongly bound to the surface. This is in contrast to gas phase thermochemistry, where formation energies would generally be expected to scale with the molecule size. Using a non-Bayesian approach like linear regression would also make the rest of the scheme intractable, since the uncertainty in the prediction is crucial for determining whether the intermediate should be studied in more detail or is unlikely to improve the model resolution, making a direct comparison is impossible. A linear regression combined with some correlation distance metric to predict uncertainty would probably work, but this is basically the key concept of Gaussian processes. We expect that most regression methods with estimation of uncertainty would probably do a reasonable job for this process and have included a note in the manuscript indicating this.

Reviewer 2

General Comments:

1. *I appreciate the effort of the authors in clarifying their paper. The main concern I had was the novelty of the work compared to previous work. In their rebuttal, the authors state 'Most of the cited work require significant insight into the existing model, usually using pre-existing microkinetic models and energetics.' And they continue 'We show that all of this information can be efficiently explore on-the-fly during mechanism refinement, starting from no known surface energetics.'*
 - a. We agree that most of these approaches are established, but disagree that this work is only of interest if new regression techniques are used. Almost certainly new models or regression techniques will be developed that would make this approach incrementally faster, or more accurate in predicting transition state energies with better linear scaling relations, but these will only be incremental improvements over this work. We have already eliminated 95% of the transition state calculations, and about half of the DFT thermochemical measurements. The contribution here is that the current state-of-the-art is more than sufficient for yielding substantial improvements in mechanism refinement and guidance, so long as uncertainty at multiple levels of approximation is tracked and used to guide refinement. This paradigm is extremely attractive to practitioners of computational chemistry as it illustrates that complex new regression methods are not the barrier to addressing the complexity that we deal with on a daily basis with large chemistries.
2. *Reading the SI multiple times, I see that the mechanism is manually generated and the kinetic information relies on the scalings (Eqs. 3-5), all of which are done a priori rather than on the fly. The only aspect that is done on the fly is the thermochemistry using the GP rather than an a priori estimation scheme. The use of PCA is standard (is one of the several schemes to identify the dimensionality of the space) as is model reduction.*
 - a. The scaling relations we use in this work are extremely broad, simply either hydrogenations, or all other chemistries. The scaling relations are constructed without regard for the material type (metallic or semiconducting, single-material or alloy or oxide, etc) or for the precise active site (step, terrace, hollow, various facets, etc.). We expect that this approach would work well for most other hydrocarbon chemistries using identical scaling relations and there is no need to recreate these relations if the accuracy (about 0.3 eV in the transition state energy) is sufficient. Of course, more accurate scaling relations could be used for a specific surface and reaction class, and in that case would need to be constructed ahead of time or reconstructed on the fly. Doing so would probably increase the accuracy from about 0.3 eV to about 0.1 eV uncertainty in estimated energies – significant, but probably not significant enough for a much larger reduction in computations than the work presented here.
3. *I don't see why the on the fly thermochemistry is a major advantage or significantly different from prior art. On this basis, I still believe the paper lacks the novelty for the journal and the typical breakthrough work we're used to see from this group and should be published in a specialized journal.*
 - a. Over the past decade, we have seen improvements in each of the specific methods used in this work, including thermochemistry predictions, scaling relations, and kinetic modeling. These improvements have not made their way into the daily toolbox of most computational chemists because they still remain less accurate than just doing the DFT calculations. The fundamental concept here is that these methods are already sufficiently accurate to act as a guide for computation and that tracking uncertainty throughout the process allows for estimates on the completeness of the model, which are not generally available using just one tool at a time.